# Silencing of RpATG8 impairs the biogenesis of maternal autophagosomes in vitellogenic oocytes, but does not interrupt follicular atresia in the insect vector *Rhodnius prolixus*

Jéssica Pereira[1☯], Calebe Diogo[1☯], Ariene Fonseca[1], Larissa Bomfim[1], Pedro Cardoso[1], Anna Santos[1], Uilla Dittz[1], Kildare Miranda[2], Wanderley de Souza[2], Adriana Gioda[3], Enrique R. D. Calderon[3], Luciana Araripe[4], Rafaela Bruno[4,5], Isabela Ramos[1,5]*

**1** Laboratório de Bioquímica de Insetos, Instituto de Bioquímica Médica Leopoldo de Meis Universidade Federal do Rio de Janeiro, Brazil, **2** Laboratório de Ultraestrutura Celular Hertha Meyer, Instituto de Biofísica Carlos Chagas Filho, Universidade Federal do Rio de Janeiro, Brazil, **3** Pontifícia Universidade Católica do Rio de Janeiro (PUC-Rio), Departamento de Química, Rio de Janeiro, Brazil, **4** Laboratório de Biologia Molecular de Insetos, Instituto Oswaldo Cruz, Rio de Janeiro, Brazil, **5** Instituto Nacional de Ciência e Tecnologia em Entomologia Molecular–INCT-EM/CNPq

☯ These authors contributed equally to this work.
* isabela@bioqmed.ufrj.br

## Abstract

Follicular atresia is the mechanism by which the oocyte contents are degraded during oogenesis in response to stress conditions, allowing the energetic resources stored in the developing oocytes to be reallocated to optimize female fitness. Autophagy is a conserved intracellular degradation pathway where double-membrane vesicles are formed around target organelles leading to their degradation after lysosome fusion. The autophagy-related protein 8 (ATG8) is conjugated to the autophagic membrane and has a key role in the elongation and closure of the autophagosome. Here we identified one single isoform of ATG8 in the genome of the insect vector of Chagas Disease *Rhodnius prolixus* (RpATG8) and found that it is highly expressed in the ovary during vitellogenesis. Accordingly, autophagosomes were detected in the vitellogenic oocytes, as seen by immunoblotting and electron microscopy. To test if autophagosomes were important for follicular atresia, we silenced RpATG8 and elicited atresia in vitellogenic females by Zymosan-A injections. We found that silenced females were still able to trigger the same levels of follicle atresia, and that their atretic oocytes presented a characteristic morphology, with accumulated brown aggregates. Regardless of the difference in morphology, RpATG8-silenced atretic oocytes presented the same levels of protein, TAG and PolyP, as detected in control atretic oocytes, as well as the same levels of acidification of the yolk organelles. Because follicular atresia has the ultimate goal of restoring female fitness, we tested if RpATG8-silenced atresia would result in female physiology and behavior changes. Under *insectarium* conditions, we found that atresia-induced control and RpATG8-silenced females present no changes in blood meal digestion, survival, oviposition, TAG content in the fat body, haemolymph amino acid levels and overall locomotor activity. Altogether, we found that autophagosomes are formed during oogenesis and that the silencing of RpATG8 impairs autophagosome biogenesis in the

**Data Availability Statement:** All relevant data are within the manuscript and its Supporting Information files.

**Funding:** This work was funded by the following grants: JCNE FAPERJ (E-26/2031802017; http://www.faperj.br/) and INCT-EM (16/2014; http://cnpq.br/). The funders had no role in study design, data collection and analysis, decision to publish, or preparation of the manuscript.

**Competing interests:** The authors have declared that no competing interests exist.

oocytes. Nevertheless, regarding major macromolecule degradation and adaptations to the fitness costs imposed by triggering an immune response, we found that autophagic organelles are not essential for follicle atresia in *R. prolixus*.

## Author summary

Follicular atresia is a phenomenon in response to environmental and physiological conditions in which female insects are able to signal the degeneration and resorption of their oocytes. It is crucial for the maintenance of female survival, as the energy stored in the developing oocytes can be reallocated allowing them to adapt to a stress condition. In the context of insect vectors of human diseases, such as flies, bugs and mosquitoes, the ability of the hematophagous female to interrupt oogenesis and reallocate its energy resources is strategic for safeguarding vector fitness. The cellular and molecular mechanisms that govern the oocytes degradation during atresia are mostly unknown. In this work, we found that a special degradation organelle, named autophagosome, is formed in the oocytes, and that these organelles are not needed for the oocytes to be degenerated during atresia in this insect. These findings are important in the context of vector population control as they provide us with knowledge regarding the vector's specific molecular biology. Information such as these are important, as they can be used for the elaboration and design of novel population control strategies.

## Introduction

The ability of insects to occupy almost every niche in nature and act as vectors of human and livestock diseases is due, at least in part, to their highly reproductive outputs. Some insects are able to lay a mass of eggs equivalent to half their body mass within hours. Thus, knowledge on the special molecular and cellular mechanisms of egg formation and embryo development is essential to elaborate upon novel strategies of vector population control. This is especially important for the vector borne neglected tropical diseases endemic to developing countries such as dengue fever and Chagas Disease (www.who.int/neglected_diseases/en/).

Oviparous animals, including insects, have evolved to produce female germline cells that not only enter meiosis to generate a gamete, but also differentiate into a giant cell designed to support embryo growth. To accomplish that, the oocyte accumulates macromolecules in a highly specialized cytoplasm with maternal mRNAs, proteins, ribosomes, mitochondria and a set of specialized endocytic-originated vesicles named yolk organelles, which usually occupy more than 99% of the mature oocyte cytoplasm [1–3]. After fertilization, clearance of maternal mRNA that were important for oogenesis and programmed degradation of the contents stored in the yolk organelles are crucial to support the anabolic metabolism of the growing embryo and, therefore, to allow successful development [4]. Thus, along with the yolk, the oocyte also accumulates a set of degradation enzymes, which are only activated after fertilization. The signals that activate programmed and selected degradation of specific components at early embryogenesis are mostly unidentified. It has been described that specific yolk organelles are acidified by the action of proton pumps such as $H^+$-ATPases [5,6] and $H^+$-PPases [7], leading to the activation of hydrolytic enzymes which target the yolk proteins [8,9]. These findings led to the description of the yolk organelles in the oocytes as "sleeping lysosomes" [6,10].

Degeneration of the ovarian follicles (follicular atresia), with major degradation of follicle contents, can occur prematurely during the time course of oogenesis [11–16]. Follicle atresia is

important to adjustments of the female to environmental and physiological conditions such as nutritional status, mating status, host deprivation and infectious processes, among others, allowing the energetic resources stored in the developing oocytes to be reallocated to optimize the female fitness [15,17,18]. In this regard, it is known that immune defense imposes fitness costs on invertebrate hosts, triggering follicle atresia, as has been well established in malaria-mosquito systems [12,14,19,20]. In Hemiptera, follicular atresia has been investigated in *Rhodnius prolixus* and *Dipetalogaster maxima*, both Chagas Disease vectors. In *R. prolixus*, follicle atresia is triggered by the immune response elicited by the direct injection of a non-pathogenic fungus and Zymosan-A, imposing a major arrest of oogenesis [21]. *R. prolixus* atretic oocytes present organelles with the typical morphology of autophagic vacuoles, acidified yolk organelles, and an increase in hydrolase activities (which are known to be involved in the degradation of the yolk) [22]. In *D. maxima*, it was shown that apoptosis and autophagy markers are stimulated [23], and that hydrolases are activated and target the yolk during atresia [24]. In Diptera and Lepidoptera, there are evidences suggesting that follicle cells and nurse cells, in each follicle, degenerate via apoptotic and autophagic mechanisms [12–14,25]. It has been previously described that follicle atresia is accomplished by activation of the stored yolk degradation machinery, and that it constitutes an economical mechanism to reallocate energy stored as yolk content.

Autophagy is a degradation pathway of intracellular components where double-membrane vesicles, named autophagosomes, are formed around target organelles and complexes leading to their degradation after lysosome fusion, with the main purpose of recycling of macromolecules for *de novo* synthesis [26]. It is a well-conserved mechanism throughout evolution in eukaryotic cells and it is carried out by a set of conserved genes that can be found from yeast to mammals. The ATG (autophagy-related) genes have been used for the study of autophagy since their discovery starting in 1993 in *S. cerevisiae* [27]. The ATG8 protein family members are central components of the autophagy machinery. ATG8 is expressed as a cytosolic precursor and is constitutively processed by ATG4 to expose conserved glycine residues. This step is crucial for the reversible conjugation of ATG8 to phosphatidylethanolamine at the membrane of autophagosomes, via ubiquitin-like conjugation systems [28,29], and this lipidation event is essential for the autophagy pathway, being considered the main molecular marker of autophagosome formation [26].

In this work, we found that RpATG8 is highly expressed in the ovaries of vitellogenic females, and that autophagosomes are endogenously formed in vitellogenic oocytes in the insect vector of Chagas Disease *R. prolixus*. To test the role of autophagy in follicle atresia, we silenced RpATG8 and triggered atresia by Zymosan-A injections f. Interestingly, we found that the morphological characteristics of the atretic oocytes in silenced females were altered, but the overall number of atretic oocytes was not decreased. Regarding vector fitness adaptations, the lack of autophagy in the follicular atresia mechanisms did not seem to be important for the female physiology and behavior under *insectarium* conditions. We discuss that autophagosomes may participate in the degradation of minor specific targets in the oocytes, rather than contribute to the massive degradation during atresia, and the potential roles of autophagy to support clearance and yolk catabolism at early embryogenesis.

## Methods

### Ethics statement

All animal care and experimental protocols were approved by guidelines of the institutional care and use committee (Committee for Evaluation of Animal Use for Research from the Federal University of Rio de Janeiro, CEUA-UFRJ #01200.001568/2013-87, order number 155/

13), under the regulation of the national council of animal experimentation control (CON-CEA). Technicians dedicated to the animal facility conducted all aspects related to animal care under strict guidelines to ensure careful and consistent animal handling.

## Insects

Insects were maintained at a 28 ± 2˚C controlled temperature and relative humidity of 70–80%. Mated females are fed for the first time (as adult insects) in live-rabbit blood 14 to 21 days after the 5th instar nymph to adult ecdysis. After the first blood feeding, all adult insects in our insectarium are fed every 21 days. For all experiments, mated females of the second or third blood feeding were used. All animal care and experimental protocols were approved by the guidelines previously described in the ethics statement.

## Gene identification

The sequence of the RpATG8 transcript was identified from the *R. prolixus* digestive tract transcriptome database [30] (RP-1485, GAHY01001604.1) and then mapped to one single isoform of the gene *RpATG8* (RPRC014434) corresponding to the transcript RPRC014434-RA in the *Rhodnius* genome assembly (Rpro C3), all available at Vector Base (https://www.vectorbase.org/). The identification was accessed by similarity to the *Drosophila melanogaster* Atg8 sequence (NP727447.1) using tBlastn. Conserved domains were detected using the NCBI Conserved Domain Database (CDD). Alignments of the RpATG8 protein sequence to the ATG8 sequences from different species were performed using ClustalOmega.

## Extraction of RNA and cDNA synthesis

All organs were dissected 6 days after blood meal and homogenized in Trizol reagent (Invitrogen) for total RNA extraction. Reverse transcription reaction was carried out using the High Capacity cDNA Reverse Transcription Kit (Applied Biosystems), using 1 µg of total RNA after RNase-free DNase I (Invitrogen) treatment, all according to the manufacturer's protocol.

## PCR/RT-qPCR

Specific primers for RpATG8 sequence were designed to amplify a 108 bp fragment in a PCR using the following cycling parameters: 5 min at 95˚C, followed by 35 cycles of 30 s at 95˚C, 30 s at 52˚C and 30 s at 72˚C and a final extension of 15 min at 72˚C. Amplifications were observed in 2% agarose gels. Quantitative PCR (RT-qPCR) was performed in a StepOne Real-Time PCR System (Applied Biosystems) using SYBR Green PCR Master Mix (Applied Biosystems) under the following conditions: 10 min at 95˚C, followed by 40 cycles of 30 s at 95˚C and 30 s at 60˚C. RT-qPCR amplification was performed using the specific primers listed on Table 1. Rp18S and EF1 were used as endogenous controls. The relative expressions were calculated using the geometric mean of the $C_t$ (cycle threshold) obtained for both reference genes and calculated $2^{-ddCt}$, according to the minimum information for publication of quantitative Real-Time PCR experiments (MIQE) Guidelines [31]. For all RT-qPCRs the samples for each biological replicate (n = 6) were dissected from a pool of 3 insects.

## RNAi silencing

dsRNA was synthesized by MEGAScript RNAi Kit (Ambion Inc) using primers for *RpATG8* specific gene amplification with the T7 promoter sequence designed to target a region of 354 bp (Table 1). Unfed adult females were injected between the second and third thoracic segments using a 10 µl Hamilton syringe with 1 µg dsRNA (in a volume of 1 µl) and fed six days

**Table 1. Primers sequences.**

| Primer | Eficiency | Forward | Reverse |
|---|---|---|---|
| RpATG8 (RT-qPCR) | 94% | 5'-GAACAATGTAATCCCACCGACAAG-3' | 5'-CCATAGACATTTTCATCACTATACGC-3' |
| Rp18S (RT-qPCR) | 100.4% [49] | 5'-TCGGCCAACAAAAGTACACA-3' | 5'-TGTCGGTGTAACTGGCATGT-3' |
| RpEF1 (RT-qPCR) | 93% [49] | 5'-GATTCCACTGAACCGCCTTA- 3' | 5'-GCCGGGTTATATCCGATTTT-3' |
| RpAtg8 (dsRNA) | - | 5'-TAATACGACTCACTATAGGGTACTATGAAGTTTCAATATAAAGAAGAGC-3' | 5'-TAATACGACTCACTATAGGGTACTATCTTCTTCATGATGTTCCTGAT-3' |

All sequences were obtained from *Vector Base* (https://www.vectorbase.org/) and primers were synthesized by Macrogen or IDT technologies. *RpATG8* (RP-1485), *RpEF1* and Rp18S [49]. The T7 sequence is underlined.

later. Three days after the blood feeding, the knockdown efficiency was confirmed by RT-qPCR at different days, after blood meal. A fragment of 808 bp of the *E. coli MalE* gene (Gene ID: 948538) included in the control plasmid LITMUS 28iMal obtained from the HiScribe RNAi Transcription kit (New England BioLabs) was amplified by PCR using a T7 promoter-specific primer, targeting the opposing T7 promoters of the vector. The cycling conditions were: 10 min at 95˚C, followed by 35 cycles of 30 s at 95˚C, 30 s at 52˚C and 60 s at 72˚C and a final extension of 15 min at 72˚C. The amplified fragment was used as a template for the synthesis of the control dsRNA (dsMal). Adult females injected with dsRNA were fed and transferred to individual vials.

## Zymosan-A challenge

Females were injected between the second and third thoracic segments using a 10 μl Hamilton syringe 3 days after the blood feeding. 2–3 μg of Zymosan-A (Sigma-Aldrich) diluted in 1 μl of water were tested. As controls, females were injected with 1 μl water alone.

## Evaluation of survival and egg laying

All groups injected with Zymosan-A, dsRNAs and controls (10 insects per batch, 3–4 batches per treatment) were kept separately in transparent plastic jars. Mortality was recorded daily. Egg laying was recorded weekly. Additional measurements are described below. Three experiments were performed, each of them containing 8 insects per treatment (n = 24).

## Evaluation of follicle atresia and isolation of follicles

To investigate ovarian morphology and follicle development, 10 females for each treatment (n = 10) were dissected in saline 6 days after the blood meal. Their ovaries were dissected free of tracheae and ovarian sheath under the stereomicroscope. In this study, ovarian follicles were classified according to Medeiros et al, 2011 [22]. In brief, follicles were classified as healthy vitellogenic when they presented translucent homogeneous ooplasm. Follicles were considered atretic when they showed ooplasm alterations that could be identified under stereomicroscope, as described previously [32].

## Oocyte homogenates and SDS-PAGE

Vitellogenic and atretic oocytes were dissected in saline and homogenized in ice cold HEPES 50 mM pH 7.4 using a plastic pestle. Each oocyte homogenate was prepared using a pool of 3 oocytes from 3 different insects. The total amount of protein in 7 oocyte homogenates were

measured by the Lowry (Folin) method (n = 7), using as standard control 1–5 μg of BSA in a E-MAX PLUS microplate reader (Molecular devices) using SoftMax Pro 5.0 as software. 30 μg of total protein from 3 oocyte homogenates were loaded in each lane of a 12.5% SDS-PAGE (n = 3). Gels were stained with silver nitrate. Folin and BSA were obtained from Sigma-Aldrich.

## Hemolymph extraction and amino acids quantification

Hemolymphs of silenced and control females were collected 6 days after blood feeding. Once collected, the hemolymph was diluted 2x in PBS and approximately 2 mg of phenylthiourea (Sigma Aldrich). Quantitative estimation of free amino acids was performed by the ninhydrin method [33]. The hemolymph of 10 insects was tested for each treatment.

## Acidification of the yolk organelles

The yolk organelles were extracted by gently disrupting the oocytes (5 oocytes from 5 different insects per treatment, n = 5) with a plastic pestle in ice cold PBS containing 5μg/ml acridine orange (AO) (Sigma-Aldrich). After 5 min incubations in the dark, the yolk organelles suspensions were deposited on glass slides and observed at an excitation wavelength of 418 nm in a Zeiss Axioplan epifluorescence microscope equipped with a fluorescein filter set and a TK-1270 JVC color video camera.

## Production of anti-RpATG8 antibodies

Specific antibodies for the single isoform of RpATG8 were raised commercially (with GeneScript). Rabbits were immunized with a 14-amino acid peptide (NH2-MKFQYKEEHPFEKRKC-COOH) derived from the predicted N-terminal of the RpATG8 CDS obtained from the *R. prolixus* transcriptome database available at Vector Base (https://www.vectorbase.org/) (RP-1485).

## Immunoblotting

Pre-vitellogenic, vitellogenic and atretic oocytes homogenates (n = 3, each replicate obtained from a pool of 3 oocytes from 3 different insects per treatment) were separated by a 12.5% SDS-PAGE, transferred to nitrocellulose membranes and blotted using antibodies against RpATG8. Membranes were blocked in TBST (Tris 50mM, pH 7.2, NaCl 150mM, 0.1% Tween 20) containing 5% dry skimmed milk for 1h. Primary antibodies were diluted 1:2500 (RpAtg8, described above) or 1:10000 (Anti-alpha Tubulin, AbCam #ab7291) in the same buffer and incubated with the membranes for 14-16h (RpATG8) and for 1h (tubulin). The membranes were washed 3x for 5 min and then incubated with the secondary antibodies (Goat Anti-Rabbit IgG H&L HRP, AbCam #ab6721 for RpATG8 and Goat Anti-mouse IgG H&L HRP, AbCam #ab6789 for tubulin) diluted 1:3000 for 1h. After washing, the membranes were developed using the Pierce ECL Western Blotting Substrate (ThermoFisher).

## Determination of triacylglycerol (TAG) content

The abdominal fat body was dissected from control and silenced females. Each replicate (n = 4) was performed using the whole fat body from one individual insect per treatment. The organs were washed in cold 0.15 M NaCl and individually homogenized in a dounce glass tube in 200 μl of cold PBS. For the oocytes, 6 oocyte homogenates were prepared using a pool of 3 oocytes from 3 different insects (n = 6). Homogenates were then subjected to enzymatic TAG determination using Triglicérides 120 kit (Doles Reagents), according to the manufacturer's instructions.

## Determination of PolyP content

Quantification of PolyP in the atretic oocytes was done using a general protocol based on the fluorimetric analysis of the characteristic emission of the DAPI-PolyP complex, as described before by [34]. Excitation wavelength was of 420 nm and the emission wavelength was of 535 nm. $PolyP_{65}$ (Sigma-Aldrich) was used for a standard curve ranging from 90 to 540 ng of PolyP. Measurements were made using a 2030 Victor X5 fluorometer (Perkin Elmer). Each replicate (n = 4) was obtained from a pool of 3 oocytes from 3 different insects per treatment.

## Light microscopy

Vitellogenic and atretic oocytes were fixed by immersion in 4% freshly prepared formaldehyde, 2.5% glutaraldehyde in 0.1 M sodium cacodylate buffer, pH 7.3 (Electron Microscopy Sciences) for 12 h at room temperature. Samples were washed 3 times for 5 minutes in the same buffer and embedded in increasing concentrations (25%, 50%, 75% and 100%) of OCT compound medium (Tissue-TEK) plus 20% glucose as a cryoprotectant, for 12 h for each of the concentrations. Once infiltrated in pure OCT, 5 μm transversal sections of the oocytes and eggs were obtained in a cryostat. The slides were mounted in Entellan (Merck) followed by observation in a Zeiss Observer. Z1 equipped a Zeiss Axio Cam MrM operated in a differential interference contrast mode.

## Electron microscopy

For conventional transmission electron microscopy (TEM), vitellogenic oocytes were fixed in freshly prepared 4% formaldehyde, 2.5% glutaraldehyde diluted in 0.1 M sodium cacodylate buffer pH 7.3 (Electron Microscopy Sciences) at 4°C for 24 h, and then embedded in epoxy resin (Spurr) (Electron Microscopy Sciences), sectioned and stained using standard methods. For freeze fracture, the yolk organelles were obtained and processed as described in [35]. All samples were examined in a JEOL 1200 EX transmission electron microscope, operating at 80 kV.

## Locomotor activity recording

Control as well as challenged and unchallenged silenced females were individually transferred (after Zymosan injections, 4[th] day after the blood meal) to 2,5 × 15 cm glass tubes, with both ends properly sealed with nylon mesh fabric to allow respiration. Locomotor activity recording was performed with a LAM25. Activity Monitoring System (Trikinetics, Waltham, MA, USA), as previously described [36]. This system incorporates an infrared interruption method that detects movement every time the insect crosses the beam. Each movement detected by the monitor was recorded with computer software (DAMSystem3 Software). All monitors were placed in an incubator room, under a relative humidity of 75% at a constant temperature of 28˚C. Activities were recorded under an artificial photoperiodic regime of 12:12 LD (cycles of 12-h of light and 12-h of darkness) for eight days. Only insects with noticeable activity lasting for the eight days were analyzed. Activity records were summed up in intervals of 30 min and averaged among individuals across each 30 min interval for daily profile representation. At least 21 insects per treatment were tested.

## Quantification of elements by inductively coupled plasma-mass spectrometry (ICP-MS)

A volume of 100 μl of double-distilled nitric acid ($HNO_3$) 65% (v/v) (Merck) was added to the lyophilized oocyte homogenates. The acidified samples were maintained at room temperature

for 24 h. After digestion at room temperature, a hot digestion at 90 $^{o}$C for 4 h was carried out. In order to dilute the solutions, 1.3 ml of ultrapure water was added to each of the samples and their chemical composition was determined by inductively coupled plasma mass spectrometry (Nexion 300X PerkinElmer). The analytical curve and the chemical composition analysis were done using Rhodium ($^{103}$Rh) as an internal standard (IS) aiming at the reduction of non-spectral interferences and matrix effects. The samples were analyzed based on the analytical curve, blanks, and Standard Reference Material (SRM 1577a). The quantification of the elements was performed using an external analytical curve of twelve points. The Limit of Detection (LOD) and the Limit of Quantification (LOQ) determined varied from 0.01 mg L$^{-1}$ (Rb) to 92.4 mg L$^{-1}$ (Si), and 0.02 mg L$^{-1}$ (Rb) to 308.1 mg L$^{-1}$ (Si), respectively.

## Results

### RpATG8 is highly expressed in the ovaries and autophagosomes are formed in the oocytes during vitellogenesis

We first identified the sequence of RpATG8 from the *R. prolixus* digestive tract transcriptome database [30] (Gene ID RP-1485, GAHY01001604.1), and then mapped it to one single isoform of the gene RpATG8 (RPRC014434-RA) in the *Rhodnius* genome assembly (Rpro C3), with a total of 3 exons in the contig ACPB03016515.1. RpATG8 encodes a predicted protein with 117 amino acid residues with 91/84% similarity/identity with the human ATG8 (LC3). All the expected ATG8 conserved domains (GABARAP, cd01611; ATG8, PF02991; MAP, PTZ00380) were detected (S1 Fig).

Quantitative PCR (RT-qPCR) showed that the ovary and flight muscle of *R. prolixus* express an average of 3x more RpATG8 than the anterior and posterior midgut and the fat body of adult females (Fig 1A). Throughout oogenesis, RpATG8 mRNA was detected in the tropharium (structure where the germ cell cluster and the nurse cells are located) and in all stages of the developing follicles (pre-vitellogenic, vitellogenic and chorionated) (Fig 1B). Because we found high expression levels of RpATG8 in the ovaries, we next investigated the presence of autophagic organelles in the oocytes in different stages of oogenesis. Antibodies against an N-terminal sequence of RpATG8 were raised and used for immunoblotting. In the ovary, the lipidated form of RpATG8 (ATG8-II), a molecular marker of autophagic organelles formation, was absent only in pre-vitellogenic oocytes, showing that autophagosomes are formed in the oocytes during vitellogenesis (Fig 1C and 1D). As a control, we tested the immune serum in samples of a somatic tissue, the midgut epithelium dissected from vitellogenic females, and found that it labeled only the free form of RpATG8 (ATG8-I), as expected, given that autophagy is usually triggered under nutritional stress (S2 Fig). Indeed, autophagic vacuoles can be easily observed by transmission electron microscopy in the cortex of vitellogenic oocytes (Fig 2A). Double membrane vesicles, such as autophagosomes, can be detected among the yolk organelles in freeze fractured samples (Fig 2B). Also, among the varied types of vesicles and organelles present in the cortex of vitellogenic oocytes, the autophagic vacuoles were often found close to maternal mitochondria (Fig 2C and 2D).

### Silencing of RpATG8 does not impair follicular atresia, but results in atretic oocytes with a different morphology

Because we found autophagosomes among the yolk organelles, we next asked if the autophagy machinery present in the oocytes was important for degrading the follicle contents during atresia. First, we adapted the protocol from [22], in which Zymosan-A, a glucan extracted from yeast cell wall, was used to trigger an immune response and, as a consequence, follicular

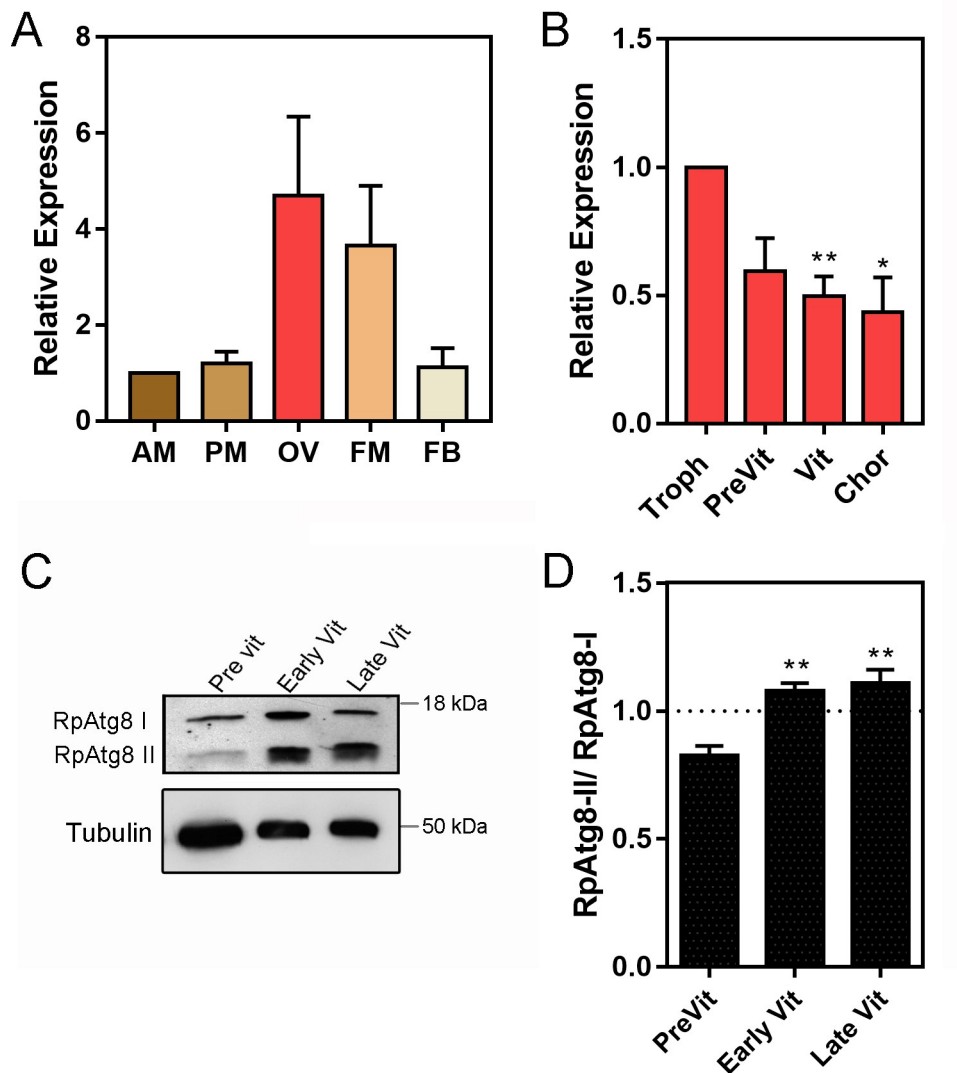

**Fig 1. RpATG8 is highly expressed in the ovaries of vitellogenic females and autophagosomes are formed during vitellogenesis in the oocytes. A.** RpATG8 mRNA quantification in the different organs of vitellogenic females dissected 7 days after the blood meal. **B.** RpATG8 mRNA quantification in the different components of the ovariole: Tropharium, pre-vitellogenic oocytes, vitellogenic oocytes and chorionated oocytes. The relative expression was quantified using the $\Delta\Delta$CT method. Graphs show mean ± SEM (n = 6). **C.** Immunoblotting using the antibodies raised against RpATG8 (LC3). Lanes 1–3: Pre vitellogenic, early vitellogenic and late vitellogenic oocytes. RpATG8-II: RpATG8 conjugated to the phosphatidylethanolamine. **D.** Immunoblotting densitometry showing the ratio of RpATG8-II/RpATG8-I (n = 3). $^*p < 0.05$, $^{**}p < 0.01$. One Way ANOVA.

atresia in *R. prolixus*. In our hands, challenging the vitellogenic females with 3,0 µg of Zymosan-A, directly injected into the hemocoel 3 days after the blood meal, resulted in the vitellogenic follicles resorption in most of the ovarioles in the ovary (Fig 3A). Next, to investigate the role of RpATG8 during follicular atresia, we synthesized a specific double-stranded RNA designed to specifically target the sequence of RpATG8 and injected it directly into the females hemocoel, 7 days before the blood meal. Quantitative PCR showed that RpATG8 knockdown was efficient, with an average of 90% mRNA silencing in the ovaries of both unchallenged (Fig 3B, Control) and challenged (Fig 3B, Zym-A) females. The bacterial *MalE* gene was used as a control dsRNA (dsMal). To test the role of the autophagy machinery in follicle atresia, we

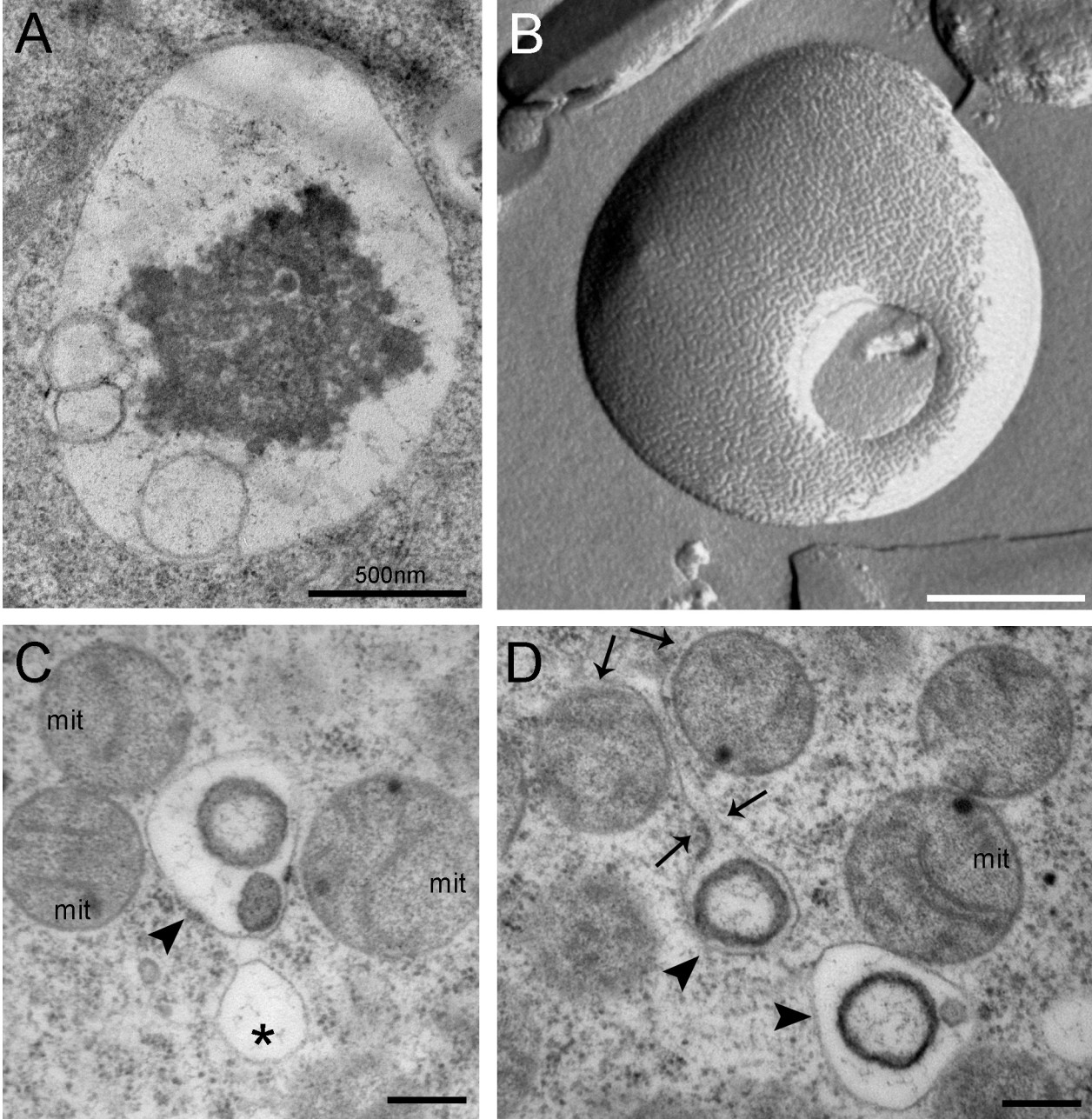

**Fig 2. Transmission electron microscopy images and freeze fracture micrographs of autophagic vacuoles in the cortex of vitellogenic oocytes.**
**A.** Autophagic vacuole observed in the peripheral cytoplasm of vitellogenic oocytes. **B.** Autophagic vacuole observed after freeze fracturing. Autophagosome delimited by a double membrane. **C-D.** Organelles observed in the cortex of chorionated (mature) oocytes. Asterisks label empty vacuoles. Arrowheads point to autophagic organelles. Arrows point to the prolongation of an organelle membrane, apparently enclosing mitochondria. m, mitochondria. Bars: 500 nm.

quantified the number of atretic oocytes elicited by Zymosan-A injections in control (dsMal) and RpATG8 (dsATG8) silenced females and found that silencing of RpATG8 does not alter the overall number of atretic follicles (Fig 3C). Despite the similar levels of atresia, interestingly, most of the atretic oocytes found in silenced females presented an altered morphology when compared with the atretic oocytes found in control females (Fig 3C). RpATG8-silenced

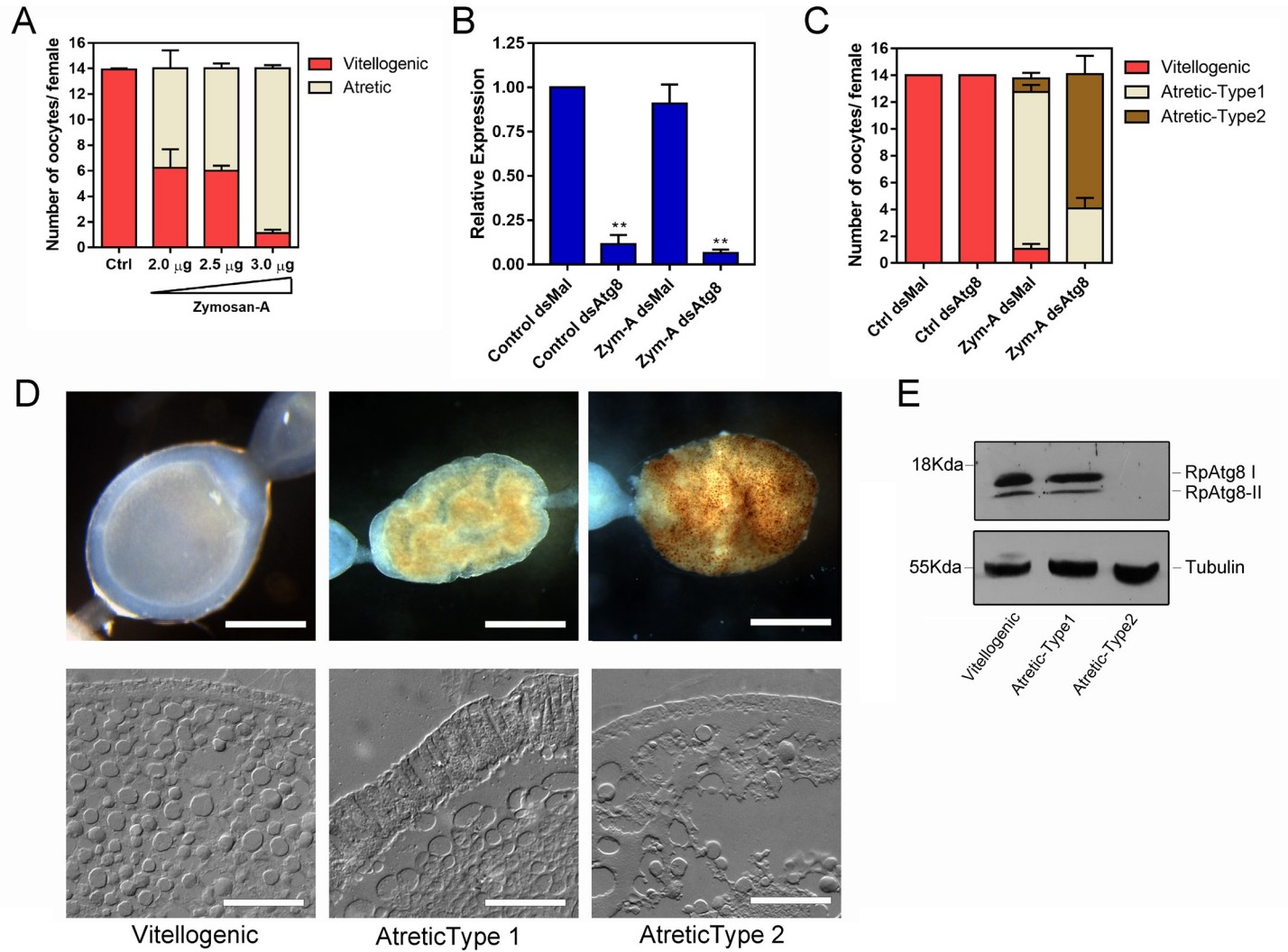

**Fig 3. Silencing of RpATG8 results in the same number of atretic oocytes, but with a different morphology. A.** Increasing concentrations of Zymosan-A were directly injected in the hemocoel of 10 vitellogenic females per treatment 3 days after the blood meal and the number of atretic oocytes was accessed 7 days after the blood meal. Graph shows mean ± SEM (n = 10). **B.** Levels of RpATG8 mRNA silencing in control and Zymosan-A-challenged females, 7 days after the blood meal. dsMal: control dsRNA, dsATG8: dsRNA designed to specifically target the RpATG8 sequence. Graph shows mean ± SEM (n = 6). **p<0.01, ***p<0.001, One Way ANOVA. **C.** The challenge with Zymosan-A was performed in control and silenced females and the number and types of atretic oocytes were accessed in 10 insects per treatment. Graph shows mean ± SEM (n = 10). **D.** Both types of atretic oocytes were observed under the stereomicroscope (Upper panel, Bars: 0.5 mm) and cryosections were observed in the light microscope operating in differential interferential contrast mode (Lower panel). Bars: 500 μm. **E.** Immunoblotting to test the silencing of RpATG8 in control and atretic oocytes. Tubulin was used as loading control (n = 3).

atretic oocytes, which we named "Type-2 atretic", present a characteristic brown punctate pattern when observed under the stereomicroscope (Fig 3D, upper panel). Furthermore, light microscopy of cryosections obtained from both types of atretic oocytes showed that the columnar follicular epithelium, characteristic of typical atretic oocytes (Type-1 atretic), is not formed in Type-2 atretic oocytes (Fig 3D, lower panel). To test if the mRNA knockdown resulted in reduced protein levels and autophagosome biogenesis in atretic oocytes, we performed immunoblotting and found that the levels of both free (RpATG8-I) and lipidated RpATG8 (RpATG8-II) were markedly decreased in silenced oocytes (type-2 atretic) (Fig 3E), indicating that, indeed, autophagosome biogenesis was impaired during atresia.

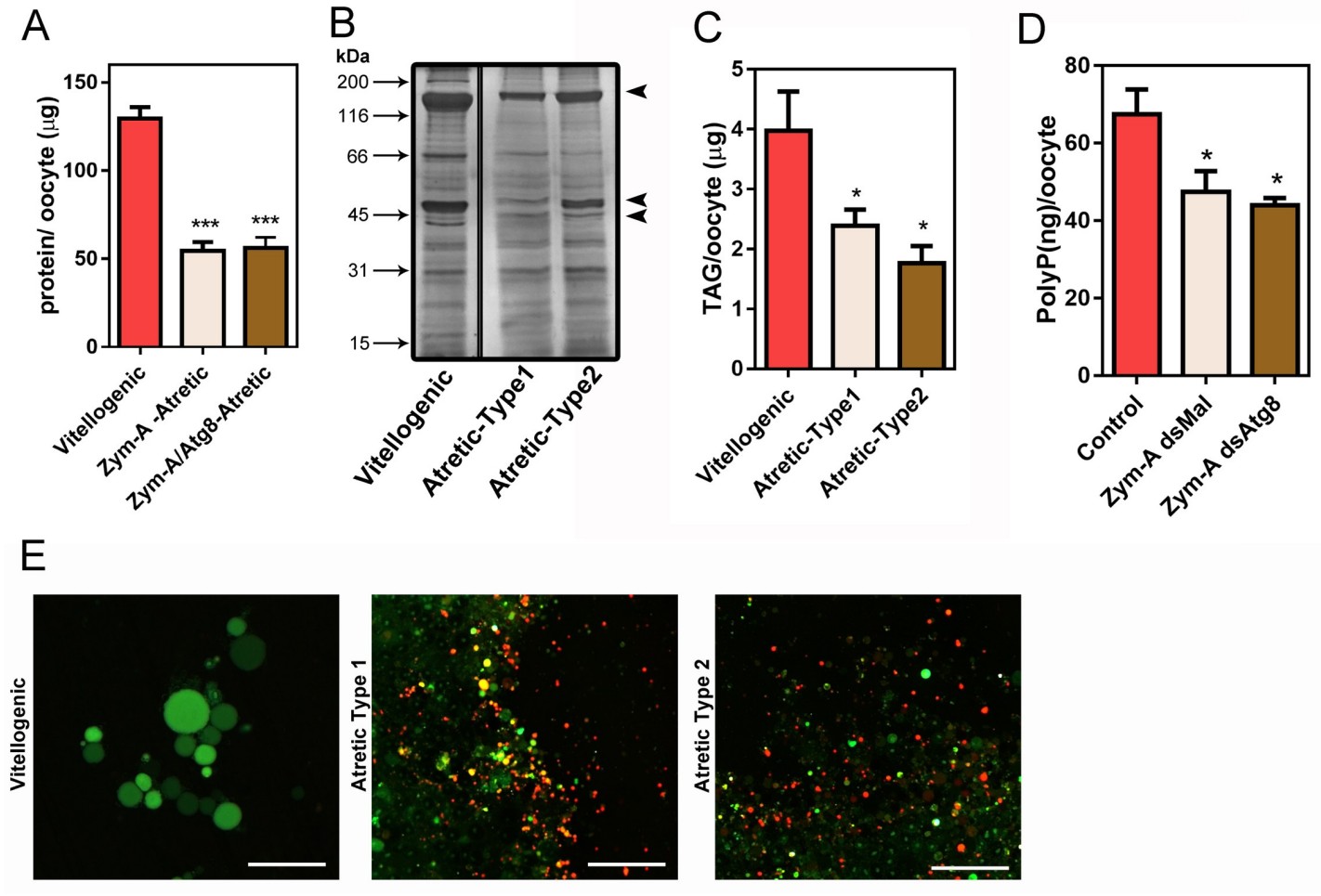

**Fig 4. Silencing of RpATG8 does not affect degradation of the main yolk macromolecules during follicular atresia. A.** Total protein quantifications in vitellogenic, control atretic (Type1) and silenced atretic (Type2) oocytes (n = 7). **B.** 10% SDS-PAGE showing the protein profile of vitellogenic and both types of atretic oocytes (n = 3). Arrows indicate vitellogenin subunits. **C.** TAG content detected in vitellogenic and both types of atretic oocytes (n = 6). **D.** PolyP content detected in vitellogenic and both types of atretic oocytes (n = 4). **E.** The yolk organelles from each of the oocytes (vitellogenic, atretic type-1 and atretic type-2) were incubated with 5μg/ml Acridine Orange (AO) and observed under the fluorescence microscope (n = 5). Bars: 50 μm. *p<0.05, ***p<0.001, One Way ANOVA.

Regardless of the differences in morphology, degradative activities in follicle atresia seem to occur similarly in control (type-1) and RpATG8-silenced (type-2) atretic oocytes, as their levels of total protein, TAG and PolyP, are approximately 60%, 40% and 30% decreased, respectively, when compared with control vitellogenic (non atretic) oocytes from the same stage (Fig 4A–4D). To test if the distinctive brownish color from type-2 atretic oocytes was the result of variations in its overall chemical composition, trace elements were quantified using ICP-MS, and no statistic differences were found between both types of atretic oocytes for their contents of iron, potassium, zinc, magnesium, calcium and copper (Table 2). Also, both types of atretic oocytes can trigger acidification of the yolk organelles during atresia, as seen by the fluorescence shift of acridine orange, a fluorescent marker of acid compartments (Fig 4E). Acidification of the yolk organelles, which culminates in the activation of yolk proteases, is characteristic of the follicular atresia degradation [37]. These data indicate that silencing of RpATG8 and impairment of autophagosome formation during vitellogenesis does not affect the major degradative pathways of follicular atresia in *R. prolixus*.

## Silencing of RpATG8 during follicular atresia does not impose major physiology costs to the female under *insectarium* conditions

As follicular atresia has the ultimate goal of restoring female fitness under stress, we asked if both types of atretic oocytes and oocyte resorption would result in comparable female physiology and behavior. Under *insectarium* conditions, we found that control and RpATG8-silenced challenged females present no changes in blood meal digestion (Fig 5A) and survival (Fig 5B). We also quantified free amino acid levels in the haemolymph and detected a 20% increase in its levels in challenged females, when compared to unchallenged females, probably as the result of follicle atresia. However, no changes between control and silenced challenged females were observed (Fig 5C). TAG contents in the fat body were also similarly affected in both groups of challenged females (Fig 5D). Next, we asked if the challenged females presented differences in their locomotor activity. For these experiments, Zymosan-challenged control and silenced females were monitored for their spontaneous locomotor activity for 8 days after the blood meal. As previously observed in the literature [36], *R. prolixus* nymphs present a peak of nocturnal activity in the beginning of the dark phase (Fig 5E, pink trace), which is thought to represent most of its host-seeking activity. Interestingly, we found that control adult females start an ascending locomotor activity during the day (around noon) before reaching the characteristic peak of nocturnal activity (Fig 5E, pink trace). Challenged control and silenced females, however, had markedly decreased diurnal activity, as well as a reduction of 40% and 60%, respectively, in their highest nocturnal activity (Fig 5E, pink trace: Control, gray trace: Zym-dsMal, brown trace: Zym-dsAtg8). Regarding oviposition, the challenge with Zymosan-A led to a marked decrease in the number of laid eggs, with no apparent effect resulting from the silencing of RpATG8 (Fig 5F). Despite the low levels of oviposition, the hatching rates are significantly decreased in challenged RpATG8 silenced females (Fig 5G), suggesting that RpATG8 might be important for embryonic development. Indeed, silencing of RpATG8 in unchallenged females results in 25–30% reduced hatching rates, mostly at the end of the oviposition period (Fig 6D). No apparent phenotypes were observed for the silencing of RpATG8 in unchallenged females for blood digestion, survival or oviposition (Fig 6A–6C).

## Discussion

Follicular atresia is a recurrent phenomenon in response to environmental and physiological conditions [15,17]. In insects, it is considered crucial for the maintenance of vector fitness and adaptation. Still, the mechanisms that allow the follicle contents programmed and regulated degradation, as well as the signals that trigger for the specific resorption of a few targeted oocytes are mostly unknown. Here, we found that autophagosomes, despite being an integral

**Table 2. Elemental quantification using Inductively coupled plasma mass spectrometry (ICP-MS).**

| | Sample (μg/oocyte) | | |
|---|---|---|---|
| Element | Control (n = 5) | Zym-A dsMal (n = 6) | Zym-A dsAtg8 (n = 5) |
| Potassium | 346,11 ± 70,90 | 85,98 ± 27,12 | 90,55 ± 5,12 |
| Magnesium | 32,61 ± 5,24 | 11,48 ± 2,89 | 12,94 ± 1,98 |
| Iron | 5,11 ± 1,58 | 2,54 ± 1,42 | 2,07 ± 0,44 |
| Copper | 1,27 ± 0,65 | 0,22 ± 0,11 | 0,24 ± 0,17 |
| Zinc | 11,21 ± 5,82 | 2,86 ± 1,39 | 2,44 ± 1,26 |
| Calcium | 88,36 ± 35,86 | 55,41 ± 15,37 | 66,40 ± 18,09 |

Results are expressed in micrograms/oocyte, as mean ± SEM of at least 5 independent quantifications performed in pools of 9 dissected oocytes.

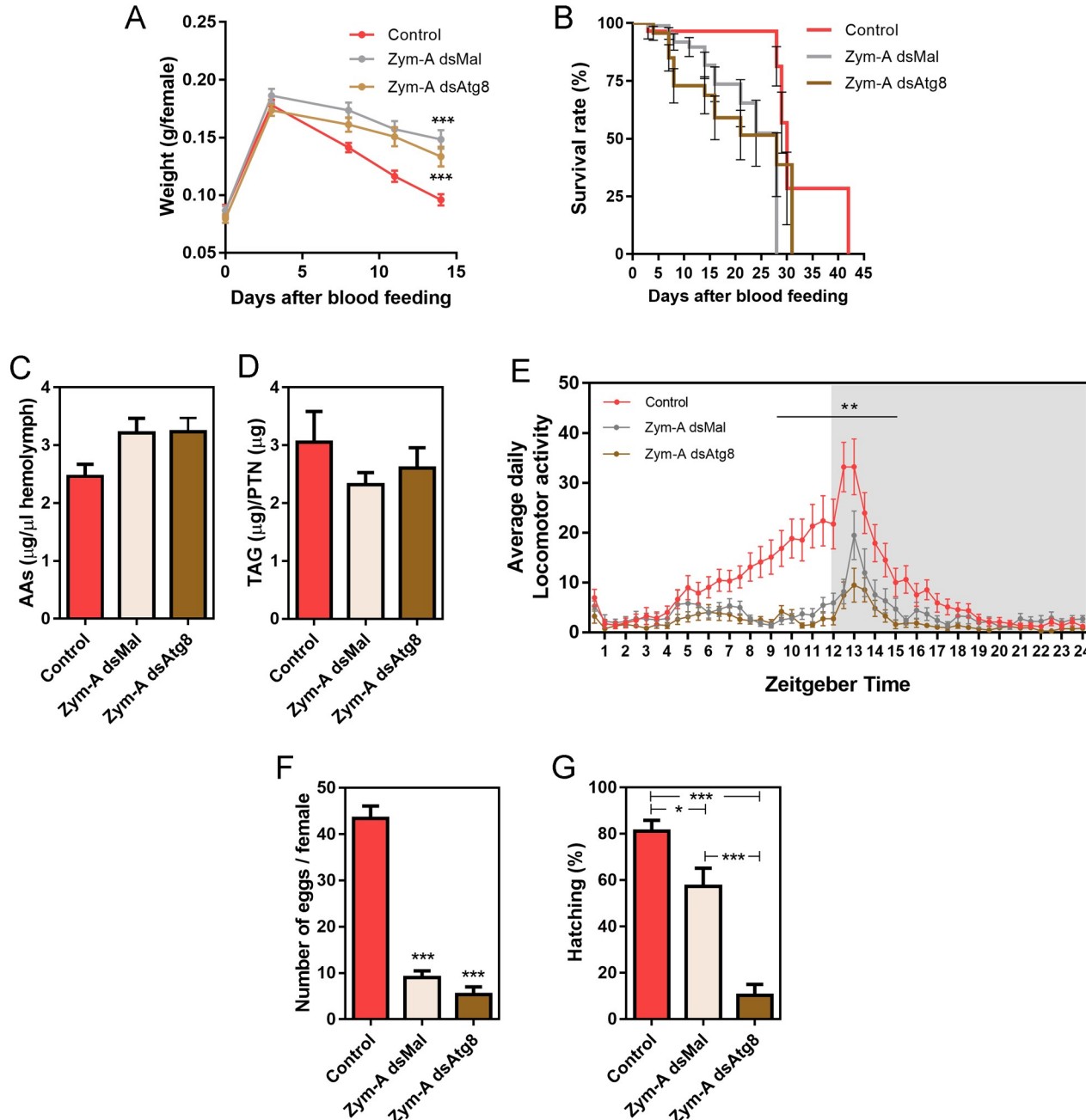

**Fig 5. Silencing of RpATG8 does not affect the physiology, longevity and locomotor behavior of vitellogenic females undergoing follicle atresia. A.** Digestion of control and challenged silenced and non-silenced females. **B.** Survival rates of control and challenged silenced and non-silenced females. For digestion and survival 3 experiments were performed. For each experiment, 8 insects per treatment were tested (n = 24). **C.** Free amino acid levels in the haemolymph (n = 10). **D.** TAG levels in the fat body (n = 4). **E.** Daily activity profile of control and challenged silenced and non-silenced females under 12:12 LD depicted by average values of eight days of recording. The gray background indicates the dark phase. Control (n = 27), Zym dsMal (n = 21), Zym dsATG8 (n = 25). **p<0.01, Two Way ANOVA. **F.** Oviposition, and **G.** Hatching rates of control and challenged silenced and non-silenced females. For oviposition and hatching, 3 experiments were performed. For each experiment, 8 insects per treatment were tested (n = 24). **p<0.01, ***p<0.001, One Way ANOVA. All measurements were performed 7 days after the blood meal. All graphs show mean ± SEM.

part of the endogenous maternally derived yolk organelles, are not essential for follicle atresia in the Hemiptera, vector of Chagas disease, *R. prolixus*.

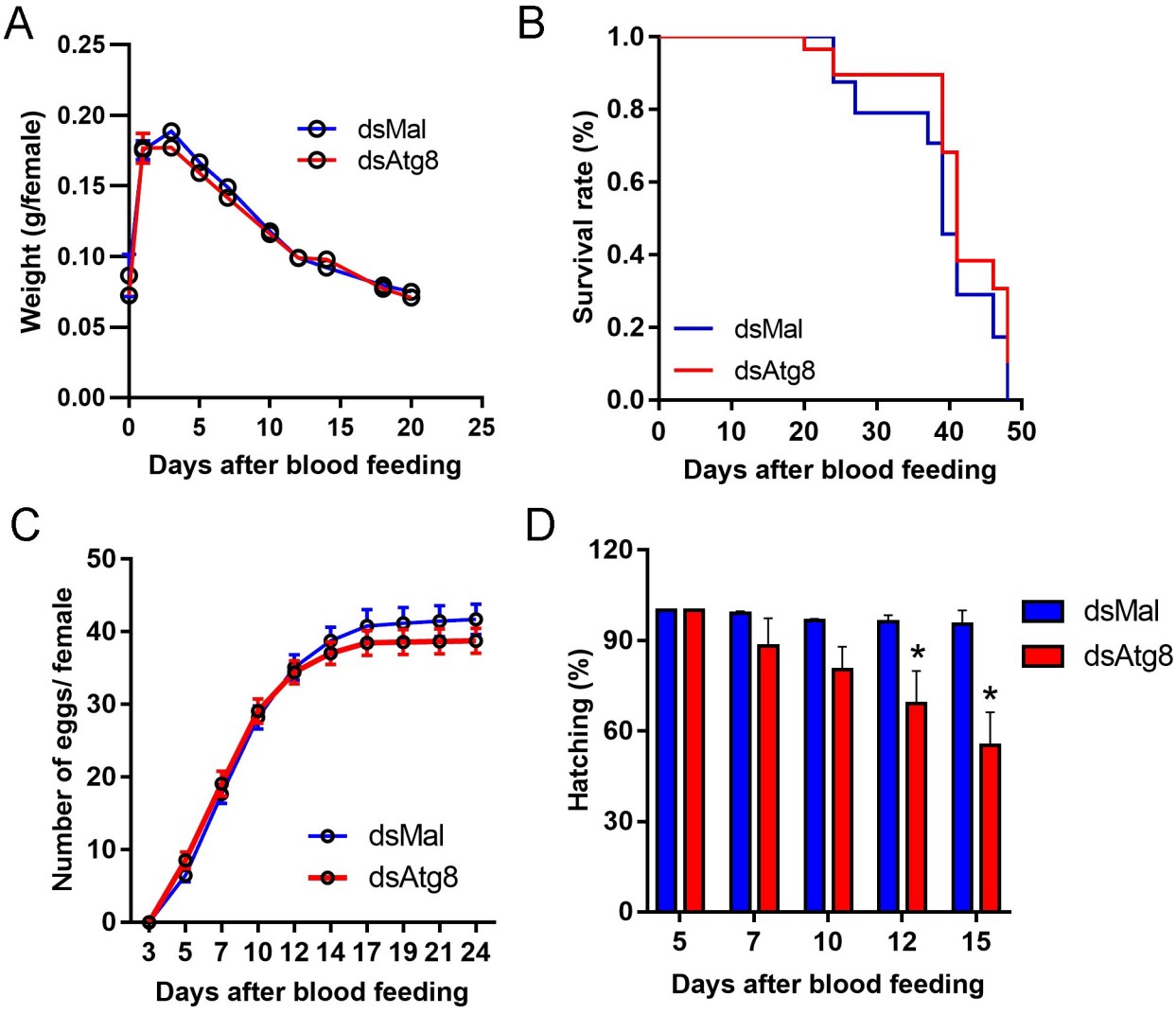

**Fig 6. Parental silencing of RpATG8 in unchallenged females leads to no changes in digestion, survival and oviposition, but results in slight reductions in embryo viability. A-B.** Digestion and longevity of control and RpATG8 silenced females. **C.** Oviposition of control and silenced females over the gonotrophic cycle. **D.** Hatching rates of the F1 from control and silenced females. Three experiments were performed. For each experiment, 8 insects per treatment were tested. Graphs show mean ± SEM (n = 24). *p<0.05, One Way ANOVA.

Most of the literature regarding molecular mechanisms of follicle atresia in insects focuses on mechanisms of programmed cell death, such as apoptosis [12–14,22,38] and autophagy [25,39,40]. In *R. prolixus*, autophagic vacuoles were observed in atretic follicles by electron microscopy [22]. In *D. maxima*, organelles with the typical morphology of autophagic vacuoles were observed as well as an increase in the lipidated form of LC3 (ATG8) during atresia [23]. This is the first report where the role of autophagy in atresia was directly tested by a gene silencing technique, and our findings show that autophagy participates, but is not essential, in the follicular atresia of *R. prolixus*. From our experiments we can conclude that the lack of autophagosomes in the oocytes does not alter their ability to go through major degeneration and resorption. We speculate that the follicles employ, and maybe, up regulate, alternative degradation pathways to allow the process of atresia to occur efficiently. Activation of pre-stored yolk degradation machinery, as previously described [22], the use of programmed cell death

degradation effectors such as caspases [41], and routes for general proteasome degradation are examples of general intracellular degradation pathways that could be important for follicle atresia. The coordinated function of these central degradation pathways is essential to cell homeostasis and adaptations to intracellular and extracellular cues that have never been tested in the specific context of follicular atresia. Still, the fact that autophagosomes are not essential for atresia does not mean that these organelles have no part in follicle degradation under control conditions. Our findings showed that silenced atretic oocytes have an evident different morphology, with accumulated brownish aggregates, indicating that the autophagic machinery does contribute to the degradation of specific components during atresia. Autophagy has been implicated in the selected degradation of ferritin and iron containing aggregates in mammalian cells [42] and silencing of heme related genes, including ferritin, results in abnormal oogenesis phenotypes [43]. Proteomics and metabolomics experiments are currently being performed in our lab aiming to determine which are the targets accumulated in silenced atretic follicles, so we can design experiments to test the specific role of autophagy in this context. Our findings also demonstrate that the silenced atretic oocytes present alterations in the morphology of the epithelial cells (when compared to control atretic oocytes) and point to the possibility that the autophagy machinery is important for this tissue to interact with the oocyte and the yolk during atresia. Changes in the morphology of the epithelial cells during atresia have been reported before in the *Culex palens* mosquito, where the epithelial cells thicken, and invaginations of this tissue enclose some of the yolk during degeneration [13]. It is important to note that because we used Zymosan-A to induce atresia, immune-response triggered melanization was observed in the hemolymph of both groups of Zymosan-challenged insects (dsMal and dsAtg8), as previously reported by Medeiros et al., 2009 [21]. However, the characteristic brown punctate pattern in the atretic oocytes was only present in Atg8 silenced females. Thus, we did not attribute this specific oocyte phenotype to the activation of the melanization cascade in the hemolymph. Additionally, it is important to mention that our dsRNA fragment targets most of the RpATG8 ORF sequence (306bp of 354bp), so there is no room in the sequence to design an additional dsRNA for a different region. Therefore, although we tested *in silico* for potential off targets with no significant predictions, it is not possible to completely rule out off target effects.

The fact that autophagosomes are an integral part of the maternally derived organelles in the oocytes points to the hypothesis that autophagy participates in the programmed degradation of specific targets that occur during early embryogenesis. In fact, silencing of RpATG8 in non-challenged females resulted in embryogenesis impairment and lower hatching rates, suggesting that the maternally derived autophagosomes have their main role throughout early development, rather than in the massive degeneration that occurs during atresia. Also, in *R. prolixus*, other components of the autophagic machinery, such as RpAtg6 [44] and RpAtg1 [personal communication] are also highly expressed in the ovary and oocytes throughout vitellogenesis, and their knockdown results in impaired embryo phenotypes as well. Activation of the autophagic flux after fertilization has already been shown in mice, where it was associated with the clearance of mRNAs for the maternal-to-zygotic transition [45]. In *C. elegans*, paternal mitochondria are degraded by autophagy after fertilization [46], and in *Drosophila*, different ATG mutants present varied phenotypes of impaired embryogenesis [47]. The maternal yolk degradation, specifically, was associated with autophagy mechanisms for the first time in 2016, in *Drosophila*. The authors show that TOR and ATG1 (autophagy-related 1) are important for yolk catabolism and the formation of autophagosomes [48].

Maternally derived autophagosomes point to a new model in which autophagy can be investigated in an endogenous and specific context. Canonically, in a typical somatic cell, autophagosome formation is triggered in response to low nutrient stress signals through the

PI3K-AKT-MTOR pathway, as a mechanism for adaptation to starvation [26]. It is, therefore, interesting that the yolk autophagosome biogenesis occurs during vitellogenesis, while the female is well fed, and the classic catabolic pathways, such as autophagy, are expected to be switched off, or running only at background levels. As oocytes are highly endocytic cells, it is possible that the biogenesis of the yolk autophagosomes is merged into the endocytic pathway, leading to the formation of transient amphisomes; still, the signals that govern recruitment of the autophagy machinery and autophagosome assembly during vitellogenesis are worth investigating, and may differ from the canonical AMPK/TOR complex nutrient sensing routes, providing evidence of new autophagy triggers. Another interesting possibility is that, like many other yolk components, the maternal autophagosomes are loaded into the oocytes at oogenesis to allow the degradation of specific targets after fertilization/egg activation, during early embryogenesis, at least before the maternal to zygotic transition. In that case, the signals that govern impairment and resumption of the autophagic flux under this peculiar endogenous context are worth investigating and may add to the literature concerning general autophagy mechanistic and function.

Altogether, we found that RpATG8 is important for the biogenesis of maternal autophagosomes in the oocytes of *R. prolixus*, and that autophagy is not essential for the mechanisms of follicular atresia in this model. We believe that these findings are important in the context of vector population, as they provide knowledge on the molecular machinery, important for oocyte formation in these animals. The identification of such molecular targets is of key importance to further understand vectors biology and to elaborate on new tactics for population control and the prevention of vector borne NTDs such as Chagas Disease.

## Supporting information

**S1 Fig. RpATG8 sequence. A.** RpATG8 sequence analysis. Gene, transcript, ORF (with the primers targeting regions) and protein conserved domains are shown. Sequence information was obtained from Vector Base (https://www.vectorbase.org/). Conserved domains were obtained from the NCBI Conserved Domains Database. PF02991 (Autophagy protein Atg8 ubiquitin like); PTZ00380 (microtubule-associated protein); CD17232 (Ubl_ATG8_GABARAP). **B.** Multiple sequence alignment of ATG8 protein sequences of different species (Clustal Omega). **C.** Matrix of similarity and identity of ATG8 protein sequences from different species (SIAS Server). Reference sequences: *Rp*, *Rhodnius prolixus;* DmAtg8, *Drosophila melanogaster* (Gene ID 42132); HsAtg8, *Homo sapiens* (Gene ID: 11337); ScAtg8, *Saccharomyces cerevisiae* (Gene ID: 852200); PaAtg8, *Periplaneta americana* (CDS GenBank: AB856588.1); BmAtg8, *Bombyx mori* (Gene ID: 692938); TmAtg8, *Tenebrio molitor* (CDS GenBank: KM676434.1). (TIF)

**S2 Fig. RpATG8 immunoblotting.** As controls, samples of the midgut and 24h-eggs were tested using the rabbit pre immune serum and RpATG8 immune serum. 45 μg of protein from each sample were used. The midgut was dissected 7 days after the blood meal. The eggs were homogenized in 50 mM HEPES, pH 7.4 20-24h after being laid by the females. The immunoblotting was performed as described in Methods. (TIF)

## Acknowledgments

The authors thank Yasmin Gutierrez, Lauriene Severiano and Desenir Pedro for maintaining the *insectarium*.

## Author Contributions

**Conceptualization:** Isabela Ramos.

**Data curation:** Jéssica Pereira, Calebe Diogo, Ariene Fonseca, Larissa Bomfim, Pedro Cardoso, Anna Santos, Uilla Dittz, Kildare Miranda, Wanderley de Souza, Adriana Gioda, Enrique R. D. Calderon, Luciana Araripe, Rafaela Bruno.

**Formal analysis:** Jéssica Pereira, Calebe Diogo, Ariene Fonseca, Larissa Bomfim, Pedro Cardoso, Anna Santos, Uilla Dittz, Adriana Gioda, Enrique R. D. Calderon, Luciana Araripe, Rafaela Bruno, Isabela Ramos.

**Funding acquisition:** Isabela Ramos.

**Investigation:** Uilla Dittz.

**Methodology:** Jéssica Pereira, Calebe Diogo, Ariene Fonseca, Larissa Bomfim, Anna Santos, Isabela Ramos.

**Project administration:** Isabela Ramos.

**Supervision:** Isabela Ramos.

**Writing – original draft:** Isabela Ramos.

**Writing – review & editing:** Jéssica Pereira, Calebe Diogo, Ariene Fonseca, Larissa Bomfim, Pedro Cardoso, Anna Santos, Uilla Dittz, Kildare Miranda, Wanderley de Souza, Luciana Araripe, Rafaela Bruno, Isabela Ramos.

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
