## [Decision Letter · Decision Letter 0]

13 Oct 2019

Dear Dr Ramos:

Thank you very much for submitting your manuscript "Silencing of RpATG8 impairs the biogenesis of maternal autophagosomes in vitellogenic oocytes but does not interrupt follicular atresia in the insect vector Rhodnius prolixus" (#PNTD-D-19-01084) for review by PLOS Neglected Tropical Diseases. Your manuscript was fully evaluated at the editorial level and by independent peer reviewers. The reviewers appreciated the attention to an important problem, but raised some substantial concerns about the manuscript as it currently stands. These issues must be addressed before we would be willing to consider a revised version of your study. We cannot, of course, promise publication at that time.

We therefore ask you to modify the manuscript according to the review recommendations before we can consider your manuscript for acceptance. Your revisions should address the specific points made by each reviewer. 

When you are ready to resubmit, please be prepared to upload the following:

(1) A letter containing a detailed list of your responses to the review comments and a description of the changes you have made in the manuscript.

(2) Two versions of the manuscript: one with either highlights or tracked changes denoting where the text has been changed (uploaded as a "Revised Article with Changes Highlighted" file); the other a clean version (uploaded as the article file).

(3) If available, a striking still image (a new image if one is available or an existing one from within your manuscript). If your manuscript is accepted for publication, this image may be featured on our website. Images should ideally be high resolution, eye-catching, single panel images; where one is available, please use 'add file' at the time of resubmission and select 'striking image' as the file type. 

Please provide a short caption, including credits, uploaded as a separate "Other" file. If your image is from someone other than yourself, please ensure that the artist has read and agreed to the terms and conditions of the Creative Commons Attribution License at http://journals.plos.org/plosntds/s/content-license (NOTE: we cannot publish copyrighted images). 

(4) If applicable, we encourage you to add a list of accession numbers/ID numbers for genes and proteins mentioned in the text (these should be listed as a paragraph at the end of the manuscript). You can supply accession numbers for any database, so long as the database is publicly accessible and stable. Examples include LocusLink and SwissProt.

(5) To enhance the reproducibility of your results, we recommend that you deposit your laboratory protocols in protocols.io, where a protocol can be assigned its own identifier (DOI) such that it can be cited independently in the future. For instructions see http://journals.plos.org/plosntds/s/submission-guidelines#loc-methods

While revising your submission, please upload your figure files to the Preflight Analysis and Conversion Engine (PACE) digital diagnostic tool, https://pacev2.apexcovantage.com/ PACE helps ensure that figures meet PLOS requirements. To use PACE, you must first register as a user. Then, login and navigate to the UPLOAD tab, where you will find detailed instructions on how to use the tool. If you encounter any issues or have any questions when using PACE, please email us at figures@plos.org.

We hope to receive your revised manuscript by Dec 12 2019 11:59PM. If you anticipate any delay in its return, we ask that you let us know the expected resubmission date by replying to this email.

To submit a revision, go to https://www.editorialmanager.com/pntd/ and log in as an Author. You will see a menu item call Submission Needing Revision. You will find your submission record there. 

Sincerely,

Joshua B. Benoit

Guest Editor

Alvaro Acosta-Serrano

Deputy Editor

I apologize for the delay in the decision as finding reviewers with expertise was exceptionally difficult. The reviewers all believe the paper is of interest , but could be improved by revision.

Reviewer's Responses to Questions

**Key Review Criteria Required for Acceptance?**

**Methods**

-Are the objectives of the study clearly articulated with a clear testable hypothesis stated?

-Is the study design appropriate to address the stated objectives?

-Is the population clearly described and appropriate for the hypothesis being tested?

-Is the sample size sufficient to ensure adequate power to address the hypothesis being tested?

-Were correct statistical analysis used to support conclusions?

-Are there concerns about ethical or regulatory requirements being met?

Reviewer #1: The analyses in this manuscript aim to understand the role of the ATG8 protein and autophagic processes more generally in the process of follicular atresia in the Kissing bug Rhodnius prolixus. Overall the experiments are well designed, controlled and represent a comprehensive analysis of the role of autophagy and ATG8 in follicular atresia. The authors utilize a variety of approaches including qPCR, western blotting, RNAi, florescent microscopy, freeze fracture electron microscopy and measurement of physiological and fitness parameters to investigate the role of ATG8 and autophagy in atresia.

Reviewer #2: The authors characterized autophagy-related protein 8 in the insect Rhodnius prolixus, and studied the effects of silencing this gene on different physiological parameters. I found several methodological weaknesses in the work. Hence, many of the results could be misinterpreted. 

-qRT-PCRs: According to the accepted standards, each qRT-PCR determination must use at least two housekeeping genes. See “Minimum Information Required for Publication of Quantitative Real-Time PCR Experiments (MIQE) Guidelines” (Bustin, S.A et al. Clin. Chem. 55, 611e622.).

-For clarity reasons, primer sequence information should be presented as a table. For primer pairs used in qRT-PCR determinations, efficiency should be informed. 

-RNAi mediated gene silencing: “using primers for RpATG6”, should be RpATG8. Detailed description of the injections is lacking: volume? method of injection? Etc. How many days after ecdysis did the females were injected? Virgin or mated females were used? Sentence: “The bacterial MalE gene was used as a control dsRNA (ref?).” Reference is lacking. Also, please specify how MalE gene was amplified: primers, template, PCR cycling conditions. Information on commercial sellers of consumables is incomplete throughout the manuscript. Importantly, off-target effects must be ruled-out with the use of another dsRNA fragment for all the determinations. 

-Bioinformatics identification of RpATG8: this result is not described in methods. If authors identified the transcript they should specify the methodology used: which gene/s was/were used as query/queries? Which Blast strategy was implemented? Please provide detailed information. 

-Immunoblotting: Provide details on the secondary antibodies used and on ECL system employed. 

-Most of the experiments have a very small sample size (n=3). This number should be increased in order to give confidence in the results.

Reviewer #3: In RNAi silencing methods chapter:

“The bacterial MalE gene was used as a control dsRNA (ref?).” Is there a missing reference? 

In Zymosan A challenge:

“As controls, females were injected with 1 µl water alone.” You have been really injecting just water? No physiological solution?

In Evaluation of survival and egg laying: 

“All groups injected with Zymosan-A, dsRNAs and controls” Specify if the group Zymosan/dsRNA is included and if control group was injected with dsRNA too.

Not exactly clear what was meant by "Mortality and egg laying were recorded daily and weekly, "respectively"."

**Results**

-Does the analysis presented match the analysis plan?

-Are the results clearly and completely presented?

-Are the figures (Tables, Images) of sufficient quality for clarity?

Reviewer #1: The results are clearly presented and explained and the figures are well designed and easy to interpret.

Reviewer #2: The results are correctlly presented, even though the methodological issues indicated above make that results difficult to interprete in many cases. I suggest to include an analysis of the sequence or RpATG8 identified (ORF, alignment with other species, conserved domains, intron-exons, etc) should be provided as supplementary information. 

-The term “silenced” for the individuals belonging to the treated group is not accurate. 

-“major organs of the adult female” is not a correct expression. The size of the organ is not related with physiological relevance. Please remove. 

-I suggest to move figure S1 to the main body of the manuscript (not supplementary information).

Reviewer #3: By qPCR you showed that the ovaries express more RpATG8 than the midgut and fat body, the other two major organs of the adult female what I consider insufficient. You should offer the whole body organ tissues check. Would be more informative to show all other organs with possible autophagy activity too.

“Type-2 atretic”, present a characteristic brown punctate pattern" is it not possible that this pattern could be caused by melanization? 

For Fig 3D the atretic type 1 is also similar to individuals treated with Zymosan A, if not could you provide the picture also for this treatment? 

For physiological and behavioral experiments do you have also results for just dsRNAi injections or they are all treated together with Zymosan A? If yes why was used combined treatment, or if you have results for just dsRNAi treatment incorporate in the charts.

For locomotor activity you provided just one sentence: "Accordingly, the overall patterns of daily locomotor activity are also similarly decreased in challenged control and silenced females." Could you please provide more extensive description of Fig 5 results?

Regarding the Fig 4E there is no explanation for this picture in whole text, so provide the description either in results section or discussion.

**Conclusions**

-Are the conclusions supported by the data presented?

-Are the limitations of analysis clearly described?

-Do the authors discuss how these data can be helpful to advance our understanding of the topic under study?

-Is public health relevance addressed?

Reviewer #1: The stated conclusions are supported by the data. I do feel that one of the more interesting findings of the analyses was placed in the supplemental materials (Figure S1). These results demonstrate a role for Atg8 and Autophagosome formation in embryonic development and egg viability. The majority of the presented results show that Atg8 is not required for the process of follicular atresia. This is informative, but I think the potential role of Atg8 in embryonic development is also worth highlighting in the context of its non-essential role in atresia. I think it would be worth considering moving Figure S1D into the main manuscript.

There was one statement in the Discussion which I don't feel is accurate and should be modified or removed.

"In the context of insect vectors of human diseases, such as flies, bugs and mosquitoes, the ability of the hematophagous female to interrupt oogenesis and reallocate the energy stored in the oocytes is strategic for safeguarding vector capacity." 

This somewhat implies that the reason for this process is to ensure that the bug maintains its vector competence. In reality this process is important for maintaining the bugs reproductive fitness by preserving deposited nutrients through resorption of partially developed oocytes. I don't believe maintenance of vector competence plays a role in the evolution of this process.

Reviewer #2: The discussion of the data presented by the authors is clear and complete.

Reviewer #3: Article conclusions are supported by the presented data and discussion includes the contribution to studied topic.

**Editorial and Data Presentation Modifications?**

Reviewer #1: I would suggest moving figure S1D to the main text and to modify the statement regarding the role of atresia in maintaining vector competence.

Reviewer #2: I suggest English edition of the manuscript.

Reviewer #3: Minor Revisions are needed according the previous comments and completion of experiment showing all tissue expression of ATG8 gene is essential.

**Summary and General Comments**

Reviewer #1: Overall, I feel this is a solid and well written paper which just requires some minor modifications prior to publication. While the results are not earth shattering, they provide some interesting insights into the relationship between autophagy and follicular atresia in an important vector species.

Reviewer #2: The manuscript address a relevant subject, and could be interesting for publication. However, the experimental and methodological problems referred above must be addressed.

Reviewer #3: (No Response)

PLOS authors have the option to publish the peer review history of their article (what does this mean?). If published, this will include your full peer review and any attached files.

Reviewer #1: No

Reviewer #2: No

Reviewer #3: No

---

## [Decision Letter · Decision Letter 1]

11 Dec 2019

Dear Dr Ramos:

Thank you very much for submitting your manuscript "Silencing of RpATG8 impairs the biogenesis of maternal autophagosomes in vitellogenic oocytes but does not interrupt follicular atresia in the insect vector Rhodnius prolixus" (PNTD-D-19-01084R1) for review by PLOS Neglected Tropical Diseases. Your manuscript was fully evaluated at the editorial level and by independent peer reviewers. The reviewers appreciated the attention to an important topic but identified some aspects of the manuscript that should be improved.

We therefore ask you to modify the manuscript according to the review recommendations before we can consider your manuscript for acceptance. Your revisions should address the specific points made by each reviewer.

(1) A letter containing a detailed list of your responses to the review comments and a description of the changes you have made in the manuscript.

(2) Two versions of the manuscript: one with either highlights or tracked changes denoting where the text has been changed (uploaded as a "Revised Article with Changes Highlighted" file ); the other a clean version (uploaded as the article file).

(3) If available, a striking still image (a new image if one is available or an existing one from within your manuscript). If your manuscript is accepted for publication, this image may be featured on our website. Images should ideally be high resolution, eye-catching, single panel images; where one is available, please use 'add file' at the time of resubmission and select 'striking image' as the file type. 

Please provide a short caption, including credits, uploaded as a separate "Other" file. If your image is from someone other than yourself, please ensure that the artist has read and agreed to the terms and conditions of the Creative Commons Attribution License at http://journals.plos.org/plosntds/s/content-license (NOTE: we cannot publish copyrighted images). 

(4) Appropriate Figure Files 

Please remove all name and figure # text from your figure files upon submitting your revision. Please also take this time to check that your figures are of high resolution, which will improve both the editorial review process and help expedite your manuscript's publication should it be accepted. Please note that figures must have been originally created at 300dpi or higher. Do not manually increase the resolution of your files. For instructions on how to properly obtain high quality images, please review our Figure Guidelines, with examples at: http://journals.plos.org/plosntds/s/figures

While revising your submission, please upload your figure files to the Preflight Analysis and Conversion Engine (PACE) digital diagnostic tool, https://pacev2.apexcovantage.com/ PACE helps ensure that figures meet PLOS requirements. To use PACE, you must first register as a user. Then, login and navigate to the UPLOAD tab, where you will find detailed instructions on how to use the tool. If you encounter any issues or have any questions when using PACE, please email us at figures@plos.org.

We hope to receive your revised manuscript by Feb 09 2020 11:59PM. If you anticipate any delay in its return, we ask that you let us know the expected resubmission date by replying to this email.

To submit your revised files, please log in to https://www.editorialmanager.com/pntd/

Sincerely,

Joshua B. Benoit

Guest Editor

Alvaro Acosta-Serrano

Deputy Editor

The manuscript is greatly improved, but there are still a few minor comments to address. The reviewers have also suggested that the manuscript should be edited for grammar and style.

Reviewer's Responses to Questions

**Key Review Criteria Required for Acceptance?**

**Methods**

-Are the objectives of the study clearly articulated with a clear testable hypothesis stated?

-Is the study design appropriate to address the stated objectives?

-Is the population clearly described and appropriate for the hypothesis being tested?

-Is the sample size sufficient to ensure adequate power to address the hypothesis being tested?

-Were correct statistical analysis used to support conclusions?

-Are there concerns about ethical or regulatory requirements being met?

Reviewer #1: (No Response)

Reviewer #2: Authors addressed the most of the Reviewer comments. I suggest minor changes:

When Vector Base is cited, include the Internet address of Vector Base.

How many days after injections the insects were fed?

Please clarify that off target effects cannot be rouled out given the reasons exposed by authors in the Reviewer responses.

**Results**

-Does the analysis presented match the analysis plan?

-Are the results clearly and completely presented?

-Are the figures (Tables, Images) of sufficient quality for clarity?

Reviewer #1: (No Response)

Reviewer #2: The results are clear and completelly presented. The information is original and relevant.

**Conclusions**

-Are the conclusions supported by the data presented?

-Are the limitations of analysis clearly described?

-Do the authors discuss how these data can be helpful to advance our understanding of the topic under study?

-Is public health relevance addressed?

Reviewer #1: (No Response)

Reviewer #2: The results are well discussed in this section.

**Editorial and Data Presentation Modifications?**

Reviewer #1: (No Response)

Reviewer #2: I suggest English edition.

**Summary and General Comments**

Reviewer #1: The authors have adequately addressed my concerns and those of the other reviewers. I feel that the paper is appropriate for publication in PLoS NTDs.

Reviewer #2: The authors addressed the Reviewers suggestions. The manuscript has been improved respect to the original version. After minor revision, I consider that the manuscript is suitable for publication in PLoS NTD.

PLOS authors have the option to publish the peer review history of their article (what does this mean?). If published, this will include your full peer review and any attached files.

Reviewer #1: No

Reviewer #2: No

---

## [Editor Report · Decision Letter 2]

23 Dec 2019

Dear Dr Ramos,

We are pleased to inform you that your manuscript, "Silencing of RpATG8 impairs the biogenesis of maternal autophagosomes in vitellogenic oocytes, but does not interrupt follicular atresia in the insect vector Rhodnius prolixus", has been editorially accepted for publication at PLOS Neglected Tropical Diseases.

Before your manuscript can be formally accepted and sent to production you will need to complete our formatting changes, which you will receive in a follow up email. Please note: your manuscript will not be scheduled for publication until you have made the required changes.

IMPORTANT NOTES

* Copyediting and Author Proofs: To ensure prompt publication, your manuscript will NOT be subject to detailed copyediting and you will NOT receive a typeset proof for review. The corresponding author will have one final opportunity to correct any errors when sent the requests mentioned above. Please review this version of your manuscript for any errors.

* If you or your institution will be preparing press materials for this manuscript, please inform our press team in advance at plosntds@plos.org. If you need to know your paper's publication date for media purposes, you must coordinate with our press team, and your manuscript will remain under a strict press embargo until the publication date and time. PLOS NTDs may choose to issue a press release for your article. If there is anything that the journal should know, please get in touch.

*Now that your manuscript has been provisionally accepted, please log into EM and update your profile. Go to http://www.editorialmanager.com/pntd, log in, and click on the "Update My Information" link at the top of the page. Please update your user information to ensure an efficient production and billing process.

*Note to LaTeX users only - Our staff will ask you to upload a TEX file in addition to the PDF before the paper can be sent to typesetting, so please carefully review our Latex Guidelines [http://www.plosntds.org/static/latexGuidelines.action] in the meantime.

Best regards,

Joshua B. Benoit

Guest Editor

Alvaro Acosta-Serrano

Deputy Editor

---

## [Editor Report · Acceptance letter]

13 Jan 2020

Dear Dr Ramos,

We are delighted to inform you that your manuscript, "Silencing of RpATG8 impairs the biogenesis of maternal autophagosomes in vitellogenic oocytes, but does not interrupt follicular atresia in the insect vector Rhodnius prolixus," has been formally accepted for publication in PLOS Neglected Tropical Diseases.

Best regards,

Serap Aksoy

Editor-in-Chief

Shaden Kamhawi

Editor-in-Chief
